# Learning Active Learning in the Batch-Mode Setup with Ensembles of Active Learning Agents

## Abstract

Supervised learning models perform best when trained on a lot of data, but annotating training data is very costly in some domains. Active learning aims to choose only the most informative subset of unlabelled samples for annotation, thus saving annotation cost. Several heuristics for choosing this subset have been developed, which use fixed policies for their choice. They are easily understandable and applied. However, there is no heuristic performing optimal in all settings. This led to the development of agents learning the best selection policy from data. They formulate active learning as a Markov decision process and apply reinforcement learning (RL) methods to it. Their advantage is that they are able to use many features and adapt to the specific task.

Our paper proposes a new approach combining these advantages of learning active learning and heuristics: We propose to learn active learning using a parameterized ensemble of agents, where the parameters are learned using Monte Carlo policy search. As this approach can incorporate any active learning agent into its ensemble, it allows to increase the performance of every active learning agent by learning how to combine it with others.

## 1 Introduction

Supervised machine learning systems perform best when trained on a large amount of training data. Obtaining this data by labelling can cause huge time and cost efforts in some domains. Active learning in the selective scenario overcomes this bottleneck by selecting a subset of all unlabelled samples to be labelled such that the model trained on them learns as much as possible and achieves a high accuracy (Cohn et al., 1994).

Heuristic active learning agents choose the samples to be labelled using a fixed policy. They have a known and predictable behaviour and their policy is easy to understand (Settles et al., 2008). However, they have two main disadvantages: First, they rarely combine different features. Second, it was found that the best heuristic highly depends on the dataset and supervised learning model used (Lowell et al., 2018).

More recently, these shortcomings have been addressed by learning active learning directly from data (Konyushkova et al., 2017; 2018; Bachman et al., 2017; Fang et al., 2017; Liu et al., 2018b;a). The authors formulate active learning as a Markov decision process and apply reinforcement learning (RL) methods like Q-Learning and imitation learning to it. While this approach promises to overcome the advantages of heuristics it introduces new problems: There is the credit assignment problem (Minsky, 1961), the training is computationally very costly (Amodei et al.), and many results are not significant and hard to reproduce (Henderson et al., 2017).

Learning active learning with RL in the batch-mode setting has only received little attention in literature. One of the reasons is that choosing a batch of samples instead of a single one makes the action space exponentially bigger and thus finding the action maximizing a value function can not be done by iterating over all actions anymore. Furthermore, it makes it harder to attribute the reward to a specific parameter of the policy.

Our paper addresses these shortcomings by proposing an active learning agent being a weighted ensemble of other agents. The weights are learned using Monte Carlo policy search and black-box optimization. As several different agents can be used as part of the ensemble, many different features can be included. By learning the weights of each agent, the ensemble can adapt to the dataset, model and optimization metric. Nonetheless, the learning is very robust and global, as black-box optimization over a small number of parameters is much easier than reinforcement learning. Furthermore, the policy is easily understandable and interpretable. The ensemble should not be seen as an alternative approach to current approaches, but rather as an extension of them allowing to learn how to combine several approaches to further increase their performance.

We evaluate our approach using active learning tasks from different domains and with random forests, CNNs and LSTMs as classifiers. The experiments show that the ensemble consistently performs at least as good as the best agent it includes and sometimes even outperforms it by a significant margin.

Our main contributions are:

- We propose an approach to learning active learning using an ensemble of heuristics. It combines the advantages of heuristics and approaches learning active learning. The experiments show that the theoretical advantages also translate into a high empirical performance.

- We show with our experiments that it is very important to train the ensemble on a similar task it is evaluated on. This is contrary to assumptions in earlier literature that a learning agent trained on a synthetic dataset works well in completely different domains.

## 2 RELATED WORK

### 2.1 ACTIVE LEARNING HEURISTICS

Heuristic frameworks are active learning frameworks relying on engineered, fixed policies. Their performance depends highly on the active learning task, with no heuristic being able to outperform the others in all cases. There are three core ideas which kind of samples should be chosen to be labelled. They can be clearly distinguished, as they rely on disjoint sets of features:

**Informativeness sampling:** One group of heuristics prefers to choose informative samples, which can be expected to change the supervised learning model a lot when added to the labelled set. Their features for a sample can be calculated only given the current supervised learning model and the sample itself. The most popular heuristic in this group is uncertainty sampling (Lewis & Gale, 1994; Scheffer et al., 2001; Shannon, 1948), while many others exist. **Diversity sampling:** This approach chooses samples which are dissimilar to already labelled samples. Its features are similarity metrics from a sample to the current labelled set. It is nearly always combined with uncertainty based sampling. **Representative sampling:** This approach, also called density-based sampling, chooses samples which are representative for the whole dataset. In particular that means that outliers should not be chosen. Like diversity sampling, it is usually combined with uncertainty sampling.

The combination of features or heuristics from these three disjoint groups is used by many researchers to increase the performance of active learning agents (Wang et al., 2017; Sener & Savarese, 2017; Zhu et al., 2009). There are many ways to combine heuristics, e.g. it is possible to first choose a subset of candidate points using one heuristic and then choose the final points to be labelled using the other heuristic, or one could directly define a heuristic combining two existing ones. While our approach with a learned ensemble is the first known approach to learn the best combination of heuristics, there exist many ideas in current literature, which could improve the performance of the proposed ensemble even further:

Wang et al. (2017) have added a diversity constraint to uncertainty sampling, and use a Quadratic Programming approach to enforce this constraint. Sener & Savarese (2017) have described active learning as a core-set-selection problem. Thus, they try to choose samples which are dissimilar both to each other and the labelled set and representative of the unlabelled set. These two approaches share that they propose advanced non-greedy methods for choosing a batch of samples. A greedy choice is only a $1 - \frac{1}{e}$-algorithm, as already pointed out by Kirsch et al. (2019). Adapting our

approach such that the samples are chosen non-greedily using the method by Wang et al. or Sener and Savarese is thus a promising future research direction.

Zhu et al. (2009) have proposed a *sampling by uncertainty and density* approach choosing the samples whose product of their uncertainty and density metric is highest. This idea can be used to extend the simple linear combination of heuristics we proposed: The product of the utility values of two heuristics or any other combination of them can easily be included as the utility of an additional heuristic of the ensemble.

Ash et al. (2019) have proposed a batch-mode AL agent choosing samples such that their gradients are diverse and used the k-means++ algorithm to choose the batch to be labelled. The diversity of gradients could be added as an additional feature for the ensemble we propose.

Active learning agents may perform better if they use different strategies in the beginning than in the end of the active learning episode. Tang & Huang (2019) have shown that it helps if the active learning agent chooses easier samples in the beginning and harder samples only in the end. This idea could also be used to extend our approach, by letting it learn a schedule of policies instead of having a constant policy.

## 2.2 LEARNING ACTIVE LEARNING

As the MDP of active learning is unknown for new tasks and datasets, methods to find the optimal policy given this MDP must rely on data. Thus, dynamic programming is not feasible, instead reinforcement learning algorithms are the suitable method for this problem. In the sequential active learning setting, the action space is a categorical one-dimensional space allowing to efficiently calculate the action maximizing a function given the action. This makes the common reinforcement learning framework of Q-Learning suitable. It fits a regression function Q(observation, action) to the training data, which tries to predict the reward given the observation for each action. Its policy $\pi(observation)$ is simply choosing the action maximizing the Q-Value given this observation. This approach is used by Konyushkova et al. (2017; 2018); Bachman et al. (2017); Fang et al. (2017). Another approach is used by Liu et al. (2018b;a): They use imitation learning and train the agent to choose the action maximizing the reward directly without estimating the reward. Their policy to imitate is a so-called algorithmic expert: It is a policy already knowing the true labels of the samples to choose and is thus able to calculate the improvement of the model if a sample is added to the labelled set. This equals evaluating the reward function without actually performing the step. They have applied this approach not only in the sequential, but also in the batch-mode active learning setting: Their agent sequentially chooses the samples to add to the batch. Our approach is also based on this idea of sequential batch-filling active learning.

The error decay can also be assumed to only depend on the number of samples in a cluster (Chang et al., 2020). They predicted the error decay using a parameterized policy depending only on this single value. Because this policy is mostly a parameterized heuristic, it only needs a small number of samples to be trained on a dataset.

Learning batch-mode active learning has also partly been covered by Ravi & Larochelle (2018): They used a Q-learning algorithm to train a regression model to predict the improvement of the accuracy if a sample is added to the labelled set. Then they multiply this quality metric with a diversity metric to gain a final expert metric. The diversity metric takes the similarity of a sample to other unlabelled samples into account. The policy is to choose the samples having the highest expert metric. While the quality metric is a learned metric, it is a design choice to maximize the product of the quality metric and the diversity metric. Thus, this approach can be seen as a mixture of an engineered heuristic and a learning agent. Their idea to use not only a combination of heuristics, but in particular a combination of a learned active learning agent and a heuristic can also be used to improve the ensemble we propose: Any learned active learning agent can be added to the set of agents making up the ensemble.

## 2.3 ENSEMBLES

In supervised learning, combinations of several different models can be combined to form an ensemble. They combine the output of these models to calculate the final output. The combination itself might be learned by another supervised learning algorithm using the predictions of the other

models as input features. This approach is called stacking and usually performs better than the best method being part of the ensemble (Wolpert, 1992). Learning the weights of an ensemble of agents can also be seen as a form of stacking, which further validates our claim that an ensemble of active learning agents performs better than the single best agent being part of it.

While ensembles of supervised learning models are widely used, there has been very little research on ensembles of agents. Nonetheless, it was found that ensembles of reinforcement learning algorithms using majority voting and Boltzmann multiplication perform significantly better than single algorithms (Wiering & Van Hasselt, 2008).

Gao et al. (2018) have proposed to learn AL by learning to choose one of three heuristics at each step of the active learning process. Different to our approach, the heuristics they used and their experiments are restricted to learning network representations. Furthermore, their approach is only applied in the sequential AL setting.

## 3    FRAMEWORK FOR LEARNING BATCH-MODE ACTIVE LEARNING

Our framework for learning batch-mode active learning is split into three parts: First, batch-mode active learning is formalized as a Markov decision process. Second, an ensemble of several different active learning is defined. Third, the parameters of this ensemble are learned using Monte Carlo policy and its objective function and optimizer are defined.

### 3.1    MARKOV DECISION PROCESS FOR SEQUENTIAL BATCH-FILLING ACTIVE LEARNING

The goal of every active learning agent is to choose the next batch of samples to be labelled based on features of the current active learning process such that the supervised learning model learns most. The MDP for pool-based active learning is described by the 4-tuple $(S, A, P, R)$:

The **state space** $S$ consists of the labelled and unlabelled set of samples and the model trained on the labelled set. As this space is extremely huge and can include millions of variables, it is very hard for an agent to learn from this state. Thus, we define an observation space, which includes features of the state which might be relevant to the active learning agent. Common metrics in this space are the predictions of a sample and metrics computed out of it, the uncertainty of the prediction and similarities or distances of unlabelled samples to each other and labelled samples. Most active learning heuristics define a single metric to rely on and do not combine them.

The **action space** $A$ is the categorical choice of one sample or a batch of samples in the unlabelled set. As the unlabelled set becomes smaller each step, the action space is variable.

Following the frameworks for learning sequential active learning (Konyushkova et al., 2017; 2018; Bachman et al., 2017; Fang et al., 2017) we are using reward shaping (Ng & Jordan, 2003) and set the reward to be the improvement of the model accuracy after a full batch is labelled and the supervised learning model is re-trained on the new labelled set. This makes the MDP a minimal extension of existing MDPs to batch-mode AL and facilitates comparison with them.

Instead of defining the **probability** $P(s'|s, a)$ of a new state given the current state and action, we define a transition function directly mapping the current state and action to the new state. In the case of active learning this step includes the annotation of a chosen sample or batch of samples, the re-training of the model with it, the change of both the labelled and unlabelled dataset, and the increase of the currently spent annotation budget.

Choosing a batch of samples to be annotated next is a very hard task: If one wants to choose $b$ samples out of a pool of $n$ unlabelled samples, there are $\binom{n}{k}$ options. Thus, we set the definition of a step of one MDP for batch-mode active learning to be the choice of one sample which is added to the batch. The corresponding pseudocode for this sequential batch-filling MDP is shown in Algorithm 1. In each step, the agent chooses the next sample to be added to the batch based on the current observation. As the observation includes similarity measures of unlabelled samples to samples in the batch, the agent can choose samples which are not too close to samples already in the batch. While the unlabelled set becomes smaller every step as one sample leaves it and enters the batch, the labelled set is only updated if the batch is full (i.e. the if-condition in the pseudocode is fulfilled): Then the labelled set is extended by all samples in the batch, thus all samples in the batch are

annotated. As the labelled set was updated, the supervised learning model can be retrained on it, changing the observation significantly.

---

**Algorithm 1** Pseudocode of the step function of a batch-mode pool-based active learning MDP

---

**function** $stepFunction(sampleToLabel)$
    $D_{unlabelled} \leftarrow D_{unlabelled} \setminus \{sampleToLabel\}$
    $batch \leftarrow batch \cup \{sampleToLabel\}$
    **if** $|batch| == batchSize$ **then**
        $D_{labelled} \leftarrow D_{labelled} \cup batch$
        $batch \leftarrow \{\}$
        $newAccuracy \leftarrow supervisedLearningModel.trainOn(D_{labelled})$
        $reward \leftarrow newAccuracy - oldAccuracy$
        $oldAccuracy \leftarrow newAccuracy$
    **else**
        $reward \leftarrow 0$
    **end if**
    $observation \leftarrow getObservation(D_{unlabelled}, D_{labelled}, batch)$
    $epochFinished \leftarrow |D_{labelled}| + |batch| >= annotationBudget$
    **return** $observation, reward, epochFinished$
**end function**

---

This definition of sequential batch-filling active learning combines the advantages of sequential and batch-mode active learning. The action per step is still one categorical choice like for sequential active learning, making it much easier for a learning agent to learn which sample to choose next. As the labelling is only performed for a full batch, the practical advantages of batch-mode learning are still preserved. However, there is a theoretical disadvantage: Compared to true sequential learning, the supervised learning model is not trained on the samples just chosen, and thus the features depending on the supervised learning model are not updated. Thus, it can be expected that a sequential batch-filling agent performs slightly worse than a true sequential agent, with the difference monotonically increasing with the batch size.

One shortcoming of this definition of the MDP is that the reward is only non-zero if a full batch was labelled. This makes reinforcement learning approaches like Q-learning difficult to apply, as it is hard to attribute the reward to the choice of a specific sample, i.e. a single action. This could be solved by re-training the supervised learning model after each sample added to the batch to calculate a fictive reward during the training of the RL approach. After this, the model has to be set back to the previous state to keep the state and observation independent of this step. Such an approach would make it possible to attribute each reward to exactly one sample, however, it has the disadvantage of needing —batch— times more re-trainings of the supervised learning model, thus increasing the computational complexity by the batch size. Hence, Monte Carlo policy search is more effective than approaches like Q-Learning.

A general challenge of learning active learning is that each computation of a reward needs the re-training of a supervised learning model, which is computationally very expensive for large models and complex problems. This property makes sample-efficiency an important criterion for choosing a suitable framework to learn active learning. Consequently, Q-Learning and Monte Carlo policy gradient are unsuited methods. Model-based RL is also unsuited, because predicting the next state is difficult for two reasons: First, the state space has multiple features for each of many samples and is thus very large. Second, predicting the next state essentially equals predicting the outcome of a neural network which is not easier than training the neural network itself.

### 3.2 WEIGHTED ENSEMBLE OF ACTIVE LEARNING AGENTS

We propose an active learning policy that is represented by an ensemble of different active learning heuristics. The basic idea is to have several different active learning agents which all assign a measure for the utility of choosing a certain data point to be labelled next and each choose the data point with the highest utility. We evaluate this approach using an ensemble of five agents:

The **random sampling** agent samples the utility of each sample independently from the same Gaussian distribution. The **uncertainty sampling** agent defines the utility of a sample as the entropy of

its prediction vector. The **diversity sampling** agent chooses samples which are dissimilar to samples already chosen for labelling. We defined its utility measure to be the minimum euclidean distance of a sample to a sample already in the labelled or batch set. The **representative sampling** agent chooses samples which are representative of the whole data distribution. In particular it aims not to choose outliers. We defined its utility as -1 * the 5%-percentile of the euclidean distance of a sample to all other samples. The **uncertainty diversity sampling** agent defines its utility as the product of the utilities of the uncertainty and diversity sampling agent (Zhu et al., 2009). The ensemble agent chooses the sample having the highest overall utility defined as the weighted sum of the utility measure of each heuristic. The corresponding function is given in Equation 1.

$$\text{utility(ensemble)} \leftarrow \sum_i \beta_i * \text{utility(agent}_i) \tag{1}$$

The beta-parameters could be chosen such that the agent performs exactly like one single heuristic by setting one $\beta$ to a high value and the others to zero. Thus, assuming this agent learned the best $\beta$-parameters for a task, its worst-case performance is the one of the best heuristic on this task, but it can also perform significantly better.

We do not claim that an ensemble made of these four simple heuristics outperforms state-of-the-art batch-mode active learning agents. We do claim, however, that the ensemble performs at least as good as all agents it contains, but can perform significantly better. Thus, any state-of-the-art active learning agent can be added to the set of agents making up the ensemble, which allows the ensemble to further improve the performance of this agent by combining it with others.

### 3.3 CHOICE OF PARAMETERS OF ENSEMBLE

The linear combination of active learning agents is a policy parameterized by the $\beta$-parameters. We decided to learn which parameters are good using Monte Carlo policy search. The objective is to maximize the performance of the supervised learning model after a complete episode. The objective function is defined in Eqn. 2 and Algorithm 2 and calculates the performance of one choice of $\beta$. Taking the mean of several active learning runs reduces the noise of the objective function. It is very hard to attribute the performance at the end of an episode to the choice of a specific action in a specific state or to the choice of $\beta$. Thus we do not solve the optimization problem using policy gradients, but treat it as a black-box function or hyperparameter optimization problem instead. Methods like Bayesian optimization, random search, grid search, tree parzen estimators, or population-based training could be used to find the optimal solution. Among them, we chose tree parzen estimators (Bergstra et al., 2013) as they are very sample-efficient and search for the global optimum.

$$\begin{aligned} &objectiveFunction(\beta) : R^n \rightarrow R \\ &with \ n \ = \ number \ of \ heuristics \ making \ up \ the \ ensemble \end{aligned} \tag{2}$$

---

**Algorithm 2** Pseudocode for the objective function of the ensemble agent

---

**Require:** $trainingTask, activeLearningEnvironment, EnsembleAgent$
  **function** $objectiveFunction(\beta)$
    $agent \leftarrow EnsembleAgent(\beta)$
    **for** $i \leftarrow 1 \ to \ numberEpisodes$ **do**
      $results \leftarrow activeLearningEnvironment.run(trainingTask, agent)$
      $performances[i] \leftarrow results.getFinalPerformance()$
    **end for**
    $meanPerformance \leftarrow mean(performances)$
    **return** $meanPerformance$
  **end function**

---

## 4 EXPERIMENTS AND RESULTS

### 4.1 SETTINGS

We compare our ensemble of five heuristics with each of these five heuristics and an ensemble not using the learned weights, but setting all weighs to the same value. We set the annotation budget such that the performance differences between the agents are clearly visible and set the batch size such that between 3 and 5 iterations could be performed. The starting set size was set to be proportional to the number of classes.

For training the $\beta$-parameters, we fixed the $\beta$-parameter for random sampling at 1 and chose a log-normal distribution with a mean of 1 and a standard deviation of 2 as prior for the other parameters.

The later parts of this section show the performance of the active learning agent on the tasks as plots of the classification accuracy over the number of labelled samples. The plots for the different evaluation tasks all share the same structure: The dark shaded area around the curves of each agent report the 95% confidence intervals, the light shaded areas are the standard deviations. A suffix to the agent names in the form of '_batchSize' denotes the batch size the agent used. There is no such suffix for the random, diversity and representative sampling agent, as their features are independent of the batch.

### 4.2 EXPERIMENTS ON UCI DATASETS

We used 11 datasets from the UCI machine learning repository (Dua & Graff, 2017). The datasets 2-breast cancer, 3-diabetes, 4-flare solar, 5-heart, 6-german, 7-mushrooms, 8-waveform, 9-wdbc were used for training and 0-adult, 1-australian, and 10-spam for evaluation, similar to Konyushkova et al. (2018). The objective function was set to be the harmonic mean of the final accuracies the ensemble reached on each of the 7 training tasks. This allows training on multiple datasets easily and thus improves generalization. The results for the 1-australian evaluation task is shown in Figure 1, the results for the other evaluation tasks are quite similar.

### 4.3 EXPERIMENTS ON CHECKERBOARD DATASETS

The datasets checkerboard 2x2 and rotated checkerboard 2x2 were used as training tasks to train the learning agents, the checkerboard 4x4 task was used as evaluation task. All checkerboard tasks are XOR-like binary classification tasks with two-dimensional input data. We used a random forest as a classifier and the results are shown in Fig. 1.

### 4.4 EXPERIMENTS ON (FASHION-) MNIST TASKS

The image classification tasks use the MNIST (LeCun et al., 2010) and fashion-MNIST (Xiao et al., 2017) dataset and a three-layer convolutional neural network implemented in keras (Chollet et al., 2015a) as classifier. Because Munjal et al. (2020) have shown that performance differences between different active learning agents may vanish when more regularization techniques are used, we tried to maximize the performance of the random baseline using following techniques: random data augmentation, dropout, L1- and L2 regularization and hyperparameter optimization using tree parzen estimators (Bergstra et al., 2013). We did not use more advanced networks like the ResNet or VGG16 because they are computationally much more expensive and tend to overfit their huge number of parameters on the very small training set size. Instead of calculating the distance metrics using the raw images, we used tSNE embeddings (Maaten & Hinton, 2008). We trained the ensemble agent on the MNIST task and evaluated it on the fashion-MNIST task, the results are shown in Fig. 1.

### 4.5 EXPERIMENTS ON QUESTION ANSWERING TASKS

The bAbI dataset (Weston et al., 2015) is a collection of question answering datasets of which two different challenges are used: one with a single supporting fact and one with two supporting facts. The classifier used is using a long short term memory and is taken from a keras example, which follows Sukhbaatar et al. (2015). Like for the image classification tasks, the hyperparameters were

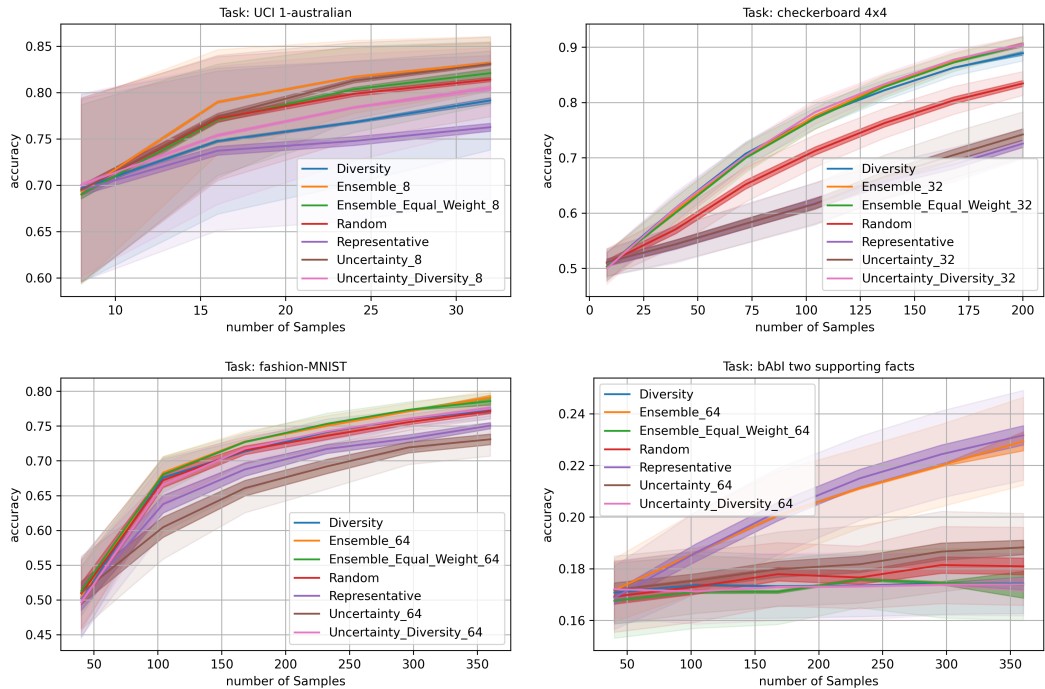

Figure 1: Results on evaluation tasks. The ensemble performs at least as good as the best heuristic, but sometimes significantly better.

optimized to minimize the loss on a small random subset. We used the bAbI task with a single supporting fact as a training task and the one with two supporting facts as an evaluation task. For calculating the distances, we used word2vec embeddings (Mikolov et al., 2015). The results are shown in Fig. 1.

## 4.6 ANALYSIS

The performance of the ensemble on the checkerboard and bAbI task was found to equal the one of the best heuristic, whereas it showed a significant performance increase compared to the best heuristic on the UCI and fashion-MNIST task. The ensemble with all weights set to 1 performed equal or worse than the one with learned weights, which shows the importance of learning the optimal weights on a training task. Our approach has also shown to have the same behaviour with respect to varying batch sizes as uncertainty sampling, see Fig. 3: It performs better with smaller batch sizes and slightly worse with larger ones. The effect of larger batch sizes depends on the weight of uncertainty and uncertainty diversity sampling, as the other two heuristics do not depend on the batch size at all.

## 4.7 INTERPRETATION OF LEARNED PARAMETERS

The ensemble learns the weights of the agents being part of it. These parameters can be easily interpreted and thus allow us to understand the ensemble's policy better. Furthermore they might be used to find out which kind of samples could be additionally generated, if possible. The weights learned for the four different tasks or tasks groups are given in Table 4.7. The weights were normalized by the sum of all weights, allowing a better comparison of the weights learned on the different tasks.

For the UCI datasets, diversity sampling is very unimportant, which is indicated by the weight of diversity sampling being nearly zero. Thus, it can be assumed that the samples in the UCI datasets are already quite dissimilar to each other. The ensemble trained on the checkerboard datasets has a weight of representative sampling being zero, which can be well explained by the fact that it does

| | random | uncertainty | diversity | representive | uncertainty diversity |
|---|---|---|---|---|---|
| UCI | 0.21 | 0.24 | 0.01 | 0.18 | 0.15 |
| checkerboard | 0.15 | 0.14 | 0.30 | 0.00 | 0.25 |
| MNIST | 0.16 | 0.08 | 0.06 | 0.28 | 0.27 |
| bAbI | 0.00 | 0.00 | 0.00 | 1.00 | 0.00 |

Table 1: Comparison of beta-parameters learned on the different training task combinations. The greener, the higher the weight.

not contain any outliers. Both MNIST and bAbI have much more samples in the unlabelled set, explaining why representative sampling is quite important for them. The weights differ a lot between the different training tasks, indicating the importance of training the parameters of the ensemble on a similar dataset it is later applied on.

## 5 Conclusion

In this paper, we proposed a novel approach for active learning using an ensemble of different active learning heuristics or trained agents. By learning the weights of a linear combination of different agents in the ensemble, this approach can be easily adapted to different datasets and supervised learning models. To train the ensemble we formalized batch-mode active learning as a Markov decision process. Experiments in different domains and using different supervised learning models have shown the effectiveness of our approach:

As expected, the worst-case performance of the ensemble is the one of the best heuristic, while it outperformed the best one significantly on some datasets. Furthermore, it was shown that it is very important to adapt the active learning agent to the dataset it is applied on.

The most natural extension of our ensemble is to make it up with more and more advanced active learning agents. Additionally, we assume following future research directions to be very promising: First, different agent weights could be learned for different time points of the episode, allowing to change the policy with increasing accuracy. Second, the weights could be learned on a small subset of the dataset it is later applied on. This allows its application on tasks where no similar training task is available or where the ensemble generalizes badly between the training and application task.

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

## A APPENDIX

### A.1 DETAILS OF CLASSIFIERS

The classifier used for both the UCI and the checkerboard datasets is the same used by Konyushkova et al. (2017): A random forest implemented with scikit-learn (Pedregosa et al., 2011) and 50 estimators. The other hyperparameters were set to the default value. We chose this classifier for easier comparison with the work by Konyushkova et al. (2017).

The classifier used for the (fashion-)MNIST task is a CNN with two convolutional layers, one intermediate dense layer and a softmax output layer. Intermediate dropout and max-pooling layers were added. The exact structure and the setting of hyperparameters can be found in the additional material in `/supervised_learning_tasks/tasks_vision/task_Vision_CNN.py:108ff`. We used random data augmentation. We had also tried the ResNet16 architecture, but found it to have worse performance, probably because its large number of parameters was overfitting on the few hundred images it was trained on. Furthermore, it was computationally much more expensive. We chose a CNN with data augmentation as classifier as it is the state-of-the-art architecture for image classification.

The classifier used for the bAbI task is a memory network (Sukhbaatar et al., 2015). The code is largely based on a keras example (Chollet et al., 2015b). We chose this architecture as it provides state-of-the-art performance.

## A.2 PERFORMANCE OF AL AGENTS WHEN TRAINED ON THE FULL DATASET

As shown in Fig 2, the worst agents need about 3 times more samples to reach the same accuracy as the best agents. They also need much longer to reach almost the same accuracy as trained on the full dataset.

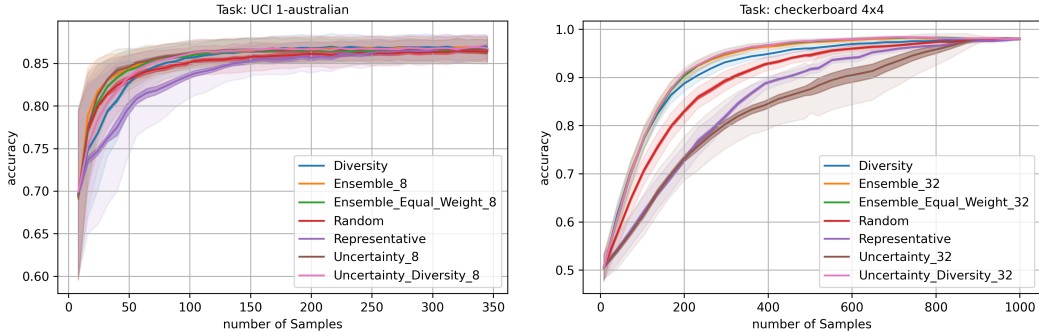

Figure 2: Results on UCI and checkerboard task with the annotation budget set to the full dataset.

## A.3 PERFORMANCE OF ENSEMBLE WITH OTHER BATCH SIZES

The plots in Fig. 3 include the ensemble with both a lower and a higher batch size than the one it was trained on. Across all task, the performance of the ensemble is monotonically decreasing with the batch size. On the UCI dataset, the ensemble performs always slightly better than uncertainty sampling with the same batch size. On the checkerboard task, the ensembles with a batch size of 2 and 32 respectively have almost the same peformance, the one with a batch size of 160 only performs slightly worse. On the fashion-MNIST task, the ensemble with a batch size of 16 performs much better than the ones with bigger batch sizes. On the bAbI task, the ensemble had learned to rely almost completely on the representative sampling agent, whose features are independent of the batch size, thus the ensemble's performance is also independent of the batch size.

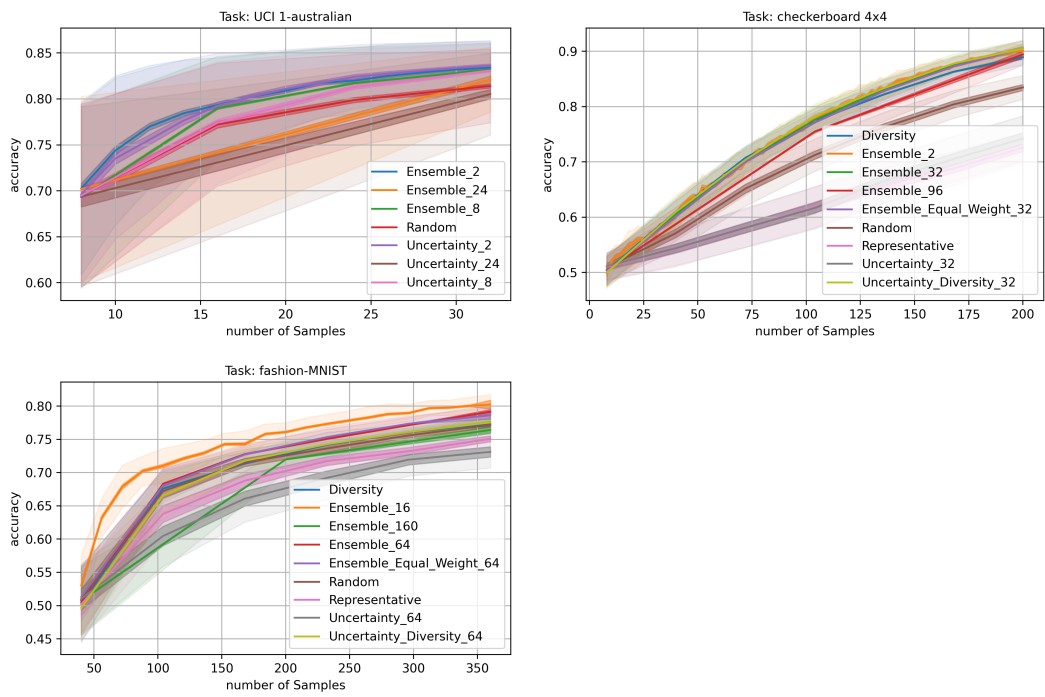

Figure 3: Results on tasks with ensemble with other batch sizes.

## A.4 DETAILS OF TRAINING TASKS

The ensemble trained on the UCI datasets 2-breast-cancer through 9-wdbc was trained with a starting size of 8, an annotation budget of 32, a batch size of 4 and the objective was set to the geometric mean of 64 episodes (8 per dataset).

The ensemble trained on the checkerboard datasets was trained with a starting size of 8, an annotation budget of 104, a batch size of 32 and the objective was set to the geometric mean of 64 episodes (32 per dataset).

The ensemble trained on the MNIST dataset was trained with a starting size of 40, an annotation budget of 296, a batch size of 64 and the objective was set to the arithmetic mean of 2 episodes.

The ensemble trained on the bAbI - single supporting fact dataset was trained with a starting size of 40, an annotation budget of 296, a batch size of 64 and the objective was set to the arithmetic mean of 2 episodes.

The starting size was set to 4 times the number of classes for each task. The annotation budget and batch size were set such that the accuracy does not converge yet and 3 to 4 batches are chosen per episode. The number of episodes run for each evaluation of the objective function was set such that the variance of the objective is approximately the same across all tasks. A geometric mean was chosen if multiple datasets were combined for training, so that the difficulty of each dataset does not influence its weight relative to the other ones. Otherwise the arithmetic mean was chosen.

## A.5 DIFFICULTY OF FINDING THE OPTIMAL BETA-PARAMETERS

The question how likely it is, that the optimal $\beta$-parameters are found, can be broken down into to subquestions:

- What does the objective function look like? The dataset, classifier and set of heuristics to choose from determine the form of the objective function mapping the $\beta$-parameters to the final accuracy of the classifier after a complete episode. Important characteristics of this function are how noisy it is and whether it contains local minima or saddle points making it

harder to find its optimum. We found for all four training tasks, that the objective function is very noisy, but does not contain any local minima.

- How likely is it, that a black-box optimizer finds the optimum of the objective function? Given the fact that only 4 continuous parameters can be optimized and the simple structure of the objective function, it can be expected that most black-box optimizers can find an optimum quite easily. However, they should be able to handle the noisy nature of the objective function.

### A.5.1 OBJECTIVE FUNCTION OF UCI TASK

The plots in Fig. 4 show the datapoints of many random evaluations of the objective function of the UCI task. The $\beta$-parameters are again normalized by their sum. The four smaller plots show the accuracy over each of the 4 $\beta$-parameters and include a fitted polynomial of second degree. It shows that high weights for uncertainty sampling should be preferred, which fits well to the fact, that the optimizer found that uncertainty sampling should be assigned the highest weight. The larger graph shows the accuracy over three of the four parameters. For all tasks, additional plots showing the accuracy over two parameters can be found in the supplementary material.

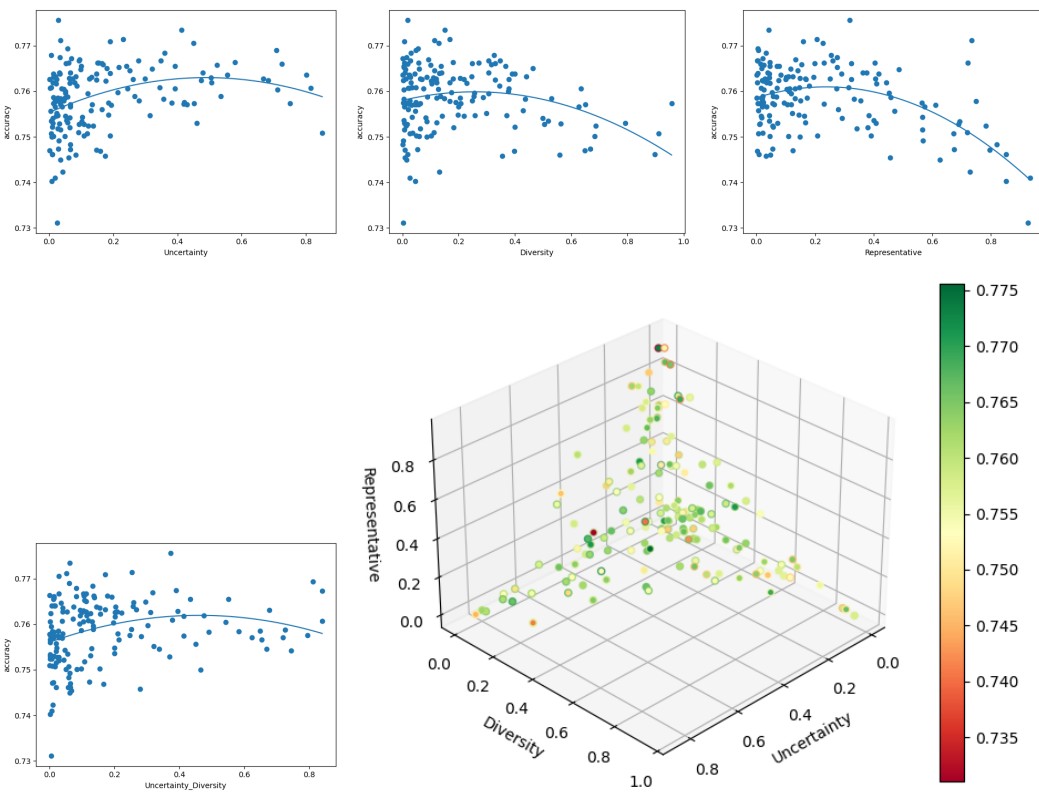

Figure 4: Objective function of the UCI task

### A.5.2 OBJECTIVE FUNCTION OF CHECKERBOARD TASKS

The plots in Fig. 5 show the datapoints of many random evaluations of the objective function of the checkerboard tasks. It is easily visible, that assigning higher weights to Diversity and Uncertainty-Diversity sampling increases the performance. This further validates that the parameters found by the optimizer are indeed good.

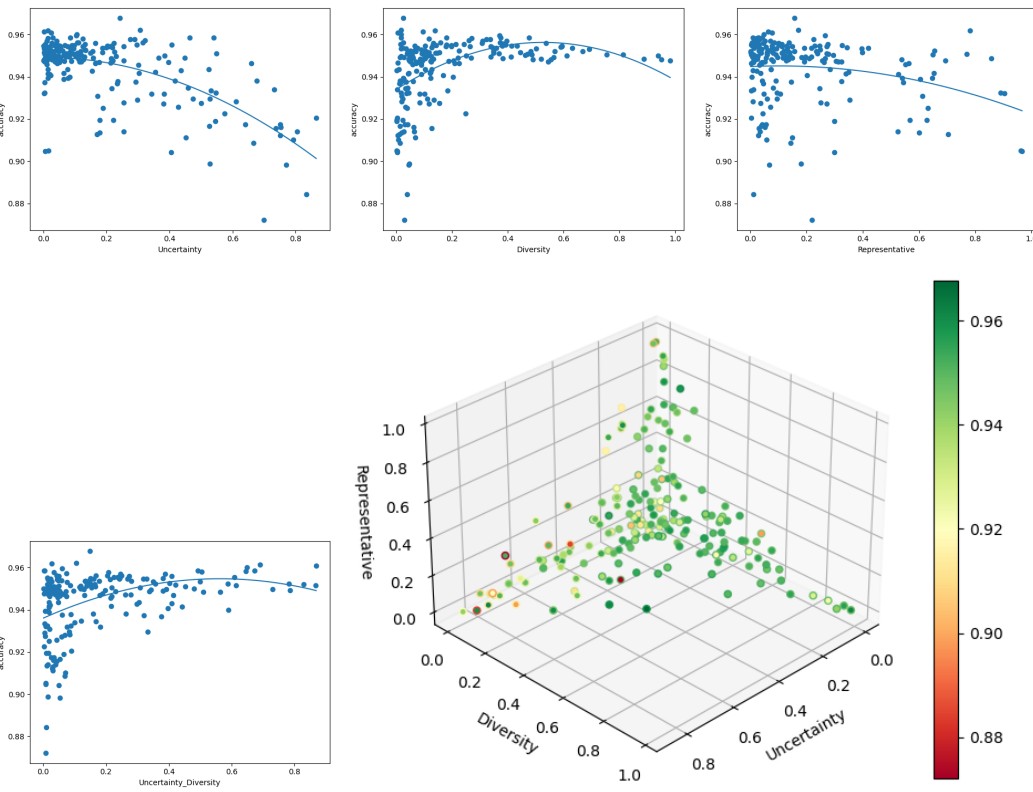

Figure 5: Objective function of the checkerboard tasks

### A.5.3    OBJECTIVE FUNCTION OF MNIST TASK

The plots in Fig. 6 show the datapoints of many random evaluations of the objective function of the MNIST task. It is clearly visible that uncertainty sampling should be assigned a low weight.

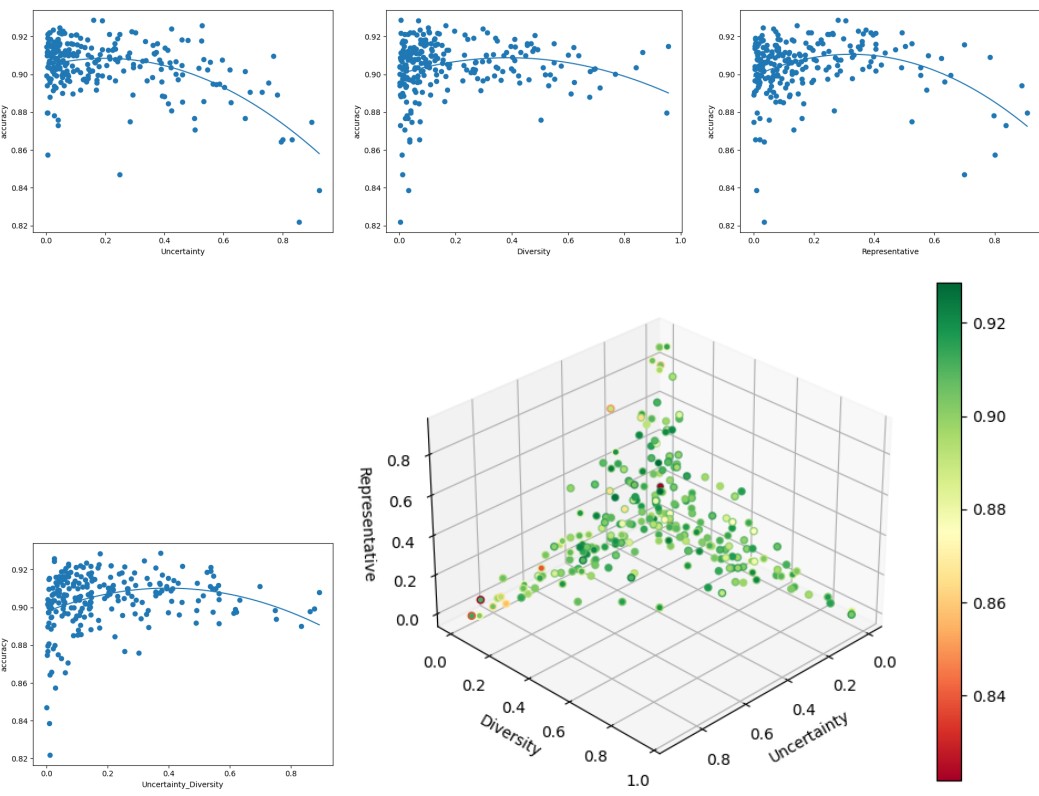

Figure 6: Objective function of the MNIST task

### A.5.4 OBJECTIVE FUNCTION OF BABI TASK

The plots in Fig. 7 show the datapoints of many random evaluations of the objective function of the bAbI task. It is clearly visible that representative sampling performs best.

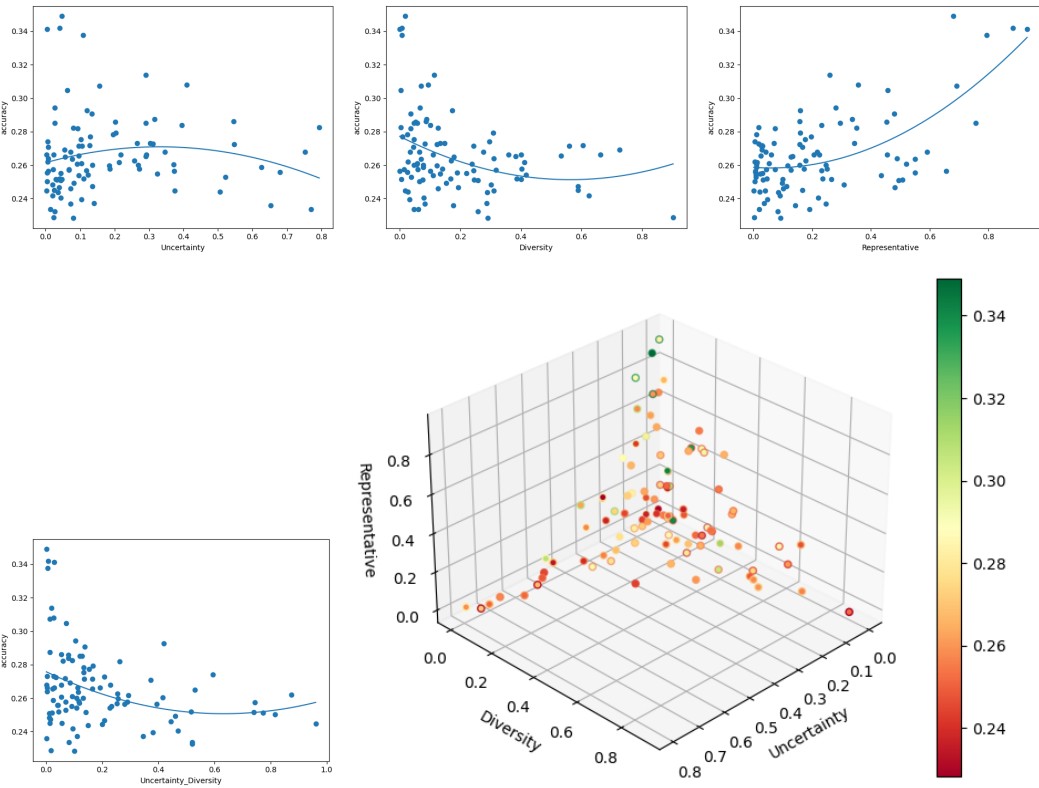

Figure 7: Objective function of the bAbI task

