# OpenReview forum: "Learning Active Learning in the Batch-Mode Setup with Ensembles of Active Learning Agents"
_ICLR.cc/2021/Conference — Reject_

### Official Review · AnonReviewer3 · 2020-10-18
**Missing comparison and connection with prior work**

**Rating:** 4
**Confidence:** 4

**Review:**

Summary of the paper:
The paper proposes an algorithm for batch-mode active learning using an ensemble of 4 active learning heuristics. The basic idea is to use an ensemble of heuristics/agents as the utility function to select a batch of samples. The paper proposes to use black-box optimization for optimizing the ratio of combining the agents. The authors perform experiments on various datasets. The results show that the proposed method outperforms the baseline heuristics in most settings, and sometimes performs significantly better.

Review:

I would vote for rejection of this paper.

1. The authors use the whole section 3.1 to describe the MDP formulation of the batch-mode active learning problem. However, the final method is not any RL algorithm but using BO. This is disappointing and also misleading. Why is policy-gradient/Q-learning not used? What will the performance be if we use RL algorithms?

2. The experiments are only comparing to the baseline heuristics, and are missing comparison with other previously proposed learning AL methods as in Section 2.2. This is not the first paper on learning to active learn in batch-mode, see e.g.:
	 Jordan T. Ash, Chicheng Zhang, Akshay Krishnamurthy, John Langford and Alekh Agarwal. Deep Batch Active Learning by Diverse, Uncertain Gradient Lower Bounds. ICLR 2020.

3. The idea of ensembling different heuristics is not new either. For example, this paper uses a multi-armed bandit view to combine heuristics:
	Gao L, Yang H, Zhou C, Wu J, Pan S, Hu Y. Active discriminative network representation learning. InIJCAI International Joint Conference on Artificial Intelligence 2018 Jan 1.

Minor Comments:

1. Last paragraph on page 4 - it should be $\binom{n}{b}$ options to choose the batch (the orders do not matter for a batch).

2. Algorithm 2: It is better to replace "noEpisodes" with "numberEpisodes".

---

> ### Author Response · Authors · 2020-11-20
> **Response to AnonReviewer3**
>
> We are very grateful for the reviewers comments and pointing out additional literature that will make our manuscript even stronger. We have addressed these points in the revised version of the manuscript.
> + “Connection to RL”
> We kindly refer to the general response.
> + “Earlier work: Deep Batch Active Learning by Diverse, Uncertain Gradient Lower Bounds”
> We appreciate pointing out the paper and included in our related work section. The flexibility of our approach would allow to include this paper into our learning framework by adding additional features for gradient diversity for the selection of a sample. Thus we believe the two approaches are orthogonal and can be combined together; rather than contradicting each other.
> + “Earlier work: Active discriminative network representation learning”
> We admit that we have missed that paper and have added it to the related work section in our revised manuscript. We have also added a description of how our work differs from theirs.
> + Minor comments
> We are thankful for them and have revised our manuscript as proposed.

---

> > ### Comment · AnonReviewer3 · 2020-11-25
> > **Thanks for the feedback**
> >
> > I've read the authors' response. Although my questions are answered, they are clearly far from being resolved;
> > 1. It is not enough to compare to RL methods conceptually. Experiments are needed to verify the intuition.
> > 2.  It is good that you cite the two papers; but I still have question about the novelty of the current paper given the two previous work, and the paper is lacking comparison to more previous works.

---

### Official Review · AnonReviewer5 · 2020-10-25
**The authors introduce a novel, batch-mode ensemble approach to Learning to Active Learn that combines the best of both worlds: heuristic- and learning- based active learning. The main idea is to create an ensemble of parametrized active learning agents that will perform better than any individual agent.**

**Rating:** 7
**Confidence:** 4

**Review:**

The authors introduce a novel, batch-mode ensemble approach to Learning to Active Learn that combines the best of both worlds: heuristic- and learning- based active learning. The main idea is to create an ensemble of parametrized active learning agents that will outperform any of the individual agents.

This is a well-written, easy to follow paper on an important topic. The work appears to be original, and the findings are significant.

The basic idea of the paper is to combine the-best-of-both-worlds of heuristic- and learning- based active learning. The authors introduce a simple & efficient way to do this. The empirical validation shows that the proposed approach is robust across a variety ML problems.

The paper would benefit by further strengthening up the empirical validation by answering the following questions:
- is the proposed approach robust wrt sample size? In other words, what happens when, for all four evaluation tasks, we consider batch sizes of 1/2/48/16/32/64. Most of these graphs could be part of an appendix, with the main paper summarizing the findings
- how many more labeled examples would take the "loser" approaches to reach the accuracy of the "winning" one(s)
- how many more iterations (and at what cost) would the proposed approach reach state-of-the-art accuracy on each evaluation task?

OTHERS:
- please run a spell-checker to avoid errors such as "acitve learning" (page 2) or "lineracombination" (page 8)

---

> ### Author Response · Authors · 2020-11-20
> **Response to AnonReviewer5**
>
> We are very grateful that the reviewer sees an interesting and valuable contribution in our paper. We are confident that we can incorporate all remarks in order to come up with an even more impactful paper.
> + “robust wrt batch size”
> We have added plots showing the performance of our approach with different batch sizes in the revised version of the manuscript, Sec. A.3. We can also give you two theoretical insights: First, only the uncertainty feature depends on the current classifier, the other two are independent of it and thus also of the batch size. Hence ensembles not relying on the uncertainty feature like the ensemble trained on the bAbI dataset have a performance independent of the batch size. Second, we found an agreement in current literature that batch-mode AL is more difficult than sequential AL and sequential heuristics perform worse in the batch-mode setting. Our experiments also found this behaviour, the performance of the ensemble is monotonically decreasing with the batch size.
> + “how many more labeled examples”
> A comparison of different approaches w.r.t. the number of samples needed to reach a certain accuracy can be given by plots showing the performance of all approaches with the annotation budget set to the size of the whole dataset. We have added such plots for the UCI and checkerboard task, see Sec. A.2 in the revised version of the manuscript,

---

### Official Review · AnonReviewer1 · 2020-10-26
**Not fully convincing**

**Rating:** 3
**Confidence:** 4

**Review:**

The paper "Learning Active Learning in the Batch-Mode Setup with Ensembles of Active Learning Agents" proposes to deal with the problem of active learning via a weighted ensemble of agents. Each agent sequentially selects data to include in the batch to be labelled according to specific heuristics. Finally, the various agents are weighted according to parameters found by a gradient-less approach.

While meta-learning of active learning is a very interesting and useful problem, I find the contribution of this paper too weak for a conference as ICLR. The approach is rather straightforward (only a linear combination of different heuristic agents), the experimental results are not fully convincing (there is no real gap between the ensemble and the best individual approach, so using the best agent at training time is maybe a strong alternative) and the paper lacks clarity and details. From my point of view the related work section should shortened to focus on mainly important aspects related to the presented work, a better detailed view (more formalized, less algorithmic) of the approach should be given, and some theoretical insights should be given before it could be considered for publication in a top machine learning conference.

---

> ### Author Response · Authors · 2020-11-20
> **Response to AnonReviewer1**
>
> We are very grateful for the reviewers comments and ideas on how to increase the contribution of our paper. We have addressed these points in the revised version of the manuscript.
> + “no real gap between the ensemble and the best individual approach”
> We agree that the ensemble only performed marginally better than the best single approach. Nonetheless, we think that such a marginal performance increase is sufficient for the following two reasons: First, the single best approach out of a set of different approaches is already a very strong benchmark. Second, we cannot know beforehand which approach performs best. Doing so would need a suitable training dataset on which all single heuristics can be tried and the average performance of multiple runs of them has to be taken. Setting up such an evaluation process needs as much effort as our approach, but our approach provides more flexibility and a better performance.
> + “The approach is pretty straightforward”
> We agree that our approach is not overly complex. Yet, at the same time it outperforms conventional approaches across several experiments and is more interpretable than other approaches building on reinforcement learning. Thus, wie believe the low complexity is a key advantage of our approach.
> + “some theoretical insights should be given”
> We think that our answers to the comments by reviewer 2 already give more insight into the approach. Nonetheless, we are happy for any proposals in which directions more theoretical insight should be provided.

---

### Official Review · AnonReviewer2 · 2020-10-28
**Review of AnonReviewer2**

**Rating:** 4
**Confidence:** 4

**Review:**

Summary:
The paper proposes learning a batch mode active learning (AL) policy as a weighted ensemble of existing AL techniques (or agents). In the proposed method, the ensemble weight vector (\beta) is learnt from data. AL is simulated on a set of training tasks where performance for various choices of \beta are estimated using a Monte Carlo approach. Subsequently, the optimal choice of \beta is found using tree Parzen estimators, a black box hyperparameter optimization technique. Experimental results are shown on Checkerboards, Fashion MNIST, bAbI and 10 UCI datasets.

+ves:
+ The Related Work section is extensive and clearly explains the existing methods in literature, building up to the reasons behind developing the proposed approach
+ The batch mode active learning framework is clearly defined and well-explained, thus setting the stage for the proposed ensemble based method.

Concerns:

- Sec 2.3, para 2, “In domains where it is not possible to learn the best parameters because there is no similar training task, an ensemble of active learning agents using such an engineered combination of them might be suitable.”
The sentence suggests that an ensemble of AL agents is suitable for domains where similar tasks are not available. However, the second contribution in Section 1 states the following. “We show with our experiments that it is very important to train the ensemble on a similar task it is evaluated on.”
This contribution seems to be in direct contradiction with the first statement.

- In Sec 3.1, the reward is chosen as the improvement in current model’s accuracy after a step i.e., choosing a sample or a batch of samples. However, Sec 3.3 and Alg 2 suggest that the objective of the \beta optimization problem is calculated as the final AL performance i.e., the performance of the supervised model at the end of the AL process. If this is the case, then reward shaping technique mentioned in Sec 3.1 might not be necessary, because the intermediate rewards are not used to learn the ensemble weights. This was not clear, kindly clarify if I missed something here.

- Sec 4.6, para 2, “..ensemble trained on the checkerboard dataset has a weight of uncertainty sampling being zero..”.
The sentence says that the weight for uncertainty sampling is zero for checkerboard, however Table 1 shows that uncertainty has weight 0.93 for the checkerboard dataset.

- The results would’ve been much stronger and more conclusive if the proposed method of learning ensemble weights had been compared against an ensemble of agents with equal weights (or random weights). Then, the superior performance of the proposed method would’ve strongly validated the need to learn ensemble weights, as opposed to giving them equal weights.

- While it is suggested in the introduction that learning a few weight parameters is much easier than reinforcement learning, the results would’ve been more complete if details were included in the Experiments section, on the computational complexity of the proposed method, as compared to RL methods in terms of time taken.

- The paper could have benefited with a more thorough discussion section. Since ensembling is a fairly well known idea, an in-depth discussion would’ve helped to understand how well ensembling works in case of active learning. For instance:
(1) Why is there a major difference of three orders of magnitude between the weights for representative sampling for MNIST and bAbI? (Table 1)
(2) In Table 1, for bAbI dataset, uncertainty and diversity have zero weight. Then, why does uncertainty-diversity sampling has a positive and high weight of 21.5?
(3) It would’ve been interesting to see an ablation study on how strongly the prior of \beta parameters would affect the final learned \beta vector.
(4) It is mentioned that if the \beta parameters are learnt properly, then the proposed method will perform at least as good as the best heuristic, in the worst case. In this vein, a discussion on how likely the tree parzen method will attain those optimal parameters, could’ve been insightful. Also, it would help the community to see the limitations of the proposed method. For instance, how do the following factors affect the optimality of \beta parameters? (a) complexity and size of the dataset (b) choice of the hyper parameter optimization technique.

- The details of what classifiers were used for the experiments on UCI datasets is not provided. A short section could have been introduced as Supplementary material to provide these details. These may be important, especially considering there seem to be different choices of classifiers for different datasets. Why was this justified, and why would this observation generalize? Some ablation studies and analysis on one dataset as to how the performance would change if another classifier was used would help showcase the effectiveness of the proposed method.

In summary, the paper aims to propose ensembling as a simple alternative to computationally costly RL techniques for active learning. While the method is well-motivated, it is missing some key experiments (especially on the significance of the contribution) and analysis, and has a weak discussion/analysis section.

Minor comments:
(which did not affect the decision):

Quite a few editorial mistakes were across the paper. Here are a few examples and their corrections. (Note that this is not an exhaustive list, and only examples of similar mistakes across the paper).
[Abstract, line 2] chose -> choose
[Abstract, line 5] fix -> fixed
[Abstract, line 8] applying -> apply
[Abstract, para 1 last sentence] “to adapt to” -> “adapt to”
[Introduction, para 3 line 1] shortcoming -> shortcomings
[Sec 2.1, para 2, line 9] which a -> which are
[Sec 2.2, para 1, line 1] dataset -> datasets
[Sec 2.3, sentence 1] “combined to an” -> combined to form an
[Sec 3, line 3] ensemble of different active learning -> ensemble of different active learning methods

POST-REBUTTAL:

I thank the authors for their response. While some of my concerns have been addressed, a few key questions haven't been answered.

* The contribution is limited in novelty.

* The general author response mentions that reward shaping is not used in the proposed method. In that case, Sec 3.1 seems a bit pedagogical and misleading, since MDP for AL is described in detail but is not even empirically compared with the proposed method.

* Regarding the ablation studies on a different classifier, while I agree that SOTA networks need to perform well, the ablation study on a different classifier was suggested to rule out the effect of the choice of SOTA classifier in the effectiveness of the proposed method. Also, the authors respond to R1 that the gap between the ensembles' performance and each single classifier's performance is small since they chose SOTA models for the individual classifiers. This is perhaps even more reason to show how ensembling works when the base models are not SOTA.

* Beyond just a statement in the response, it would have been good to see some empirical comparisons between the proposed method and, say, Q-learning-based RL methods - especially in terms of actual running time complexity.

* I also agree with R3 that similar ideas have been explored before (papers cited in R3's review), and it is important to compare with those methods as baselines in the experiments.

* Also, the choice of the datasets used is not justified appropriately.

I stay with my original decision.

---

> ### Author Response · Authors · 2020-11-20
> **Response to AnonReviewer2**
>
> We are very grateful for the reviewer's comments and the in-detail description of propositions on how to improve our paper. We have addressed these points in the revised version of the manuscript.
> + “contribution seems to be in direct contradiction”
> We agree with the reviewer, the first statement was misleading and deleted.
> + “reward shaping technique might not be necessary”
> We kindly refer to the general response.
> + “weight of uncertainty sampling being zero”
> We corrected it to ‘ weight of REPRESENTATIVE sampling’.
> + “compared against an ensemble of agents with equal weights”
> We added such an agent with all weights set to 1 to the plots. The ensemble learning the weights instead performs much better on the UCI and bAbI dataset.
> + “compared to RL methods”
> We kindly refer to the general response.
> + “Discussion of \beta parameters”
> We agree that the question which \beta parameters are found and how  likely it is that the best parameters are found is an important one. In our view, this question can be broken down into three components: i) What does the objective function of the optimization look like? ii) How does it depend on the size and complexity of the dataset? iii) How good can an optimizer find the minimum of the objective function? We have addressed i) and ii) by generating visualizations of the objective function for the different datasets and discussing properties of it like local minima and saddle points and added them to the revised version in Sec. A.5. To iii): The performance of different optimizers dependent on the qualitative structure of the objective function and the priors set for the parameters is a huge topic in itself. As the objective function only has a few parameters and no local minima, we assume that most black-box optimizers, e.g. also Bayesian Optimization, can find an optimum quite easily.
> + “Details of classifiers”
> We have added details of the classifiers and the reasoning for choosing them in the supplementary section, Sec. A.1.
> + “Ablation study on a classifier”
> Unfortunately, we did not find the time to perform these experiments. However, we think it is more important for an active learning agent to perform well with the state-of-the-art classifier for a given task (e.g. a CNN with data augmentation for image classification) than being able to generalise well across different classifiers. We chose such state-of-the-art classifiers for the MNIST and bAbI task.
> + “Editorial mistakes”
> Thank you very much for pointing them out, we have corrected them.

---

### Author Response · Authors · 2020-11-20
**General Response to all Reviewers**

We are deeply grateful for all reviewer's comments and their propositions on how to improve our paper. We have addressed their points in the revised version of the manuscript.

One point addressed by both reviewer 2 and 3 is that the MDP described includes reward shaping and thus seems to be the basis for a RL method like Q-learning. However we used a Monte Carlo method for finding the best policy, which does not need any intermediate rewards. This was found to be at least irritating.
Let us explain why we chose this definition of the MDP: We agree that reward shaping is not necessary to learn the ensemble weights. However, it was nonetheless included to allow the reader to compare our MDP to existing MDPs in the literature, like the ones used for Q-Learning (see Sec. 2.2, para 2.2). The MDP we propose is an extension of existing MDPs to batch-mode AL.

Both reviewer 2 and 3 also requested a comparison of our approach to RL methods like Q-Learning. We have revised Sec. 3.1 accordingly, here a short summary: RL-methods like Q-Learning need to receive a reward after each action taken. Thus, they need a re-training of the supervised learning model every time a sample is added to the batch. The Monte-Carlo approach, however, only needs a re-training after each batch. Thus the computational complexity of Q-Learning in settings with a batch size of b is b times higher than Monte-Carlo methods. Because of this much higher computational complexity we did not include Q-Learning approaches as baselines.
Even when using Monte-Carlo methods, running a complete active learning episode is still very expensive, as it requires multiple training of a supervised learning model. The high variance of the accuracy makes it additionally necessary to run many episodes to be able to estimate the performance of an approach with sufficient confidence. Thus sample-efficiency is a very important criterion for choosing a suitable RL approach. Monte Carlo with tree parzen estimators fulfils this criterion much better than Monte Carlo policy gradients.

---

### Decision · Program_Chairs · 2021-01-07
**Final Decision**

**Decision:**

Reject

**Comment:**

The authors propose to linearly combine the utility functions of (batch) active learning algorithms. The linear combination coefficients are "learned" with Monte Carlo estimators to adapt the coefficients to different kinds of tasks automatically.

The reviewers find the presentation within the papers generally clear. The simplicity of the approach, which is highlighted in the authors' rebuttal, should be appreciated. The authors also addressed the issue of robustness with respect to the batch size. But the paper left quite a few unanswered issues even after the authors rebuttal. The novelty with respect to several earlier papers require clarification and concrete comparisons, such as the ones in reinforcement learning and bandit learning as pointed out by reviewers. The lack of comparisons to those existing works, both illustratively and empirically, is a key weakness of the current paper. A more careful study of RL setting (such as reward shaping) is also important to understand the value of the work. Finally, the gap between the ensemble approach and the single approach also deserves more investigation to justify the significance of the contribution.